# Dietary Patterns, Not Gut Microbiome Composition, Are Associated with Behavioral Challenges in Children with Autism: An Observational Study

**DOI:** 10.3390/nu17213476

**Published:** 2025-11-04

**Authors:** Genna Di Benedetto, Germana Sorge, Marco Sarchiapone, Luca Di Martino

**Affiliations:** 1Il Cireneo SRL, 66054 Vasto, Italy; gennadibenedetto@ilcireneosrl.it (G.D.B.); g.sorge@ilcireneosrl.it (G.S.); marcosarchiapone@ilcireneosrl.it (M.S.); 2Case Digestive Health Research Institute, School of Medicine, Case Western Reserve University, Cleveland, OH 44106, USA; 3Department of Medicine, School of Medicine, Case Western Reserve University, Cleveland, OH 44106, USA

**Keywords:** autism, microbiome, mycobiome, dietary diversity

## Abstract

**Background/Objectives**: Autism Spectrum Disorder (ASD) is a complex neurodevelopmental condition characterized by persistent social communication difficulties and restricted, repetitive behaviors, with prevalence estimates continuing to rise worldwide. The gut–brain axis has been proposed as a potential contributor to ASD, yet human studies yield inconsistent findings, partly due to confounding effects of diet and behavior. **Methods**: Here, we investigated the gut bacteriome and mycobiome of children with ASD (*n* = 17) compared with their non-ASD siblings (*n* = 9) and parents without ASD (*n* = 27), alongside detailed assessment of dietary intake (*n* = 79) using 7-day food diaries. **Results**: Multi-kingdom microbiome profiling revealed no significant differences in α- or β- diversity across ASD, sibling, and parental groups, with only minor taxonomic variation observed. Similarly, fungal community composition showed negligible group-level differences. By contrast, dietary patterns strongly differentiated ASD from non-ASD participants: children with ASD consumed higher levels of sweets and sugary foods, lower portions of vegetables, and exhibited reduced overall dietary diversity. Statistical analyses confirmed that dietary factors, rather than microbial composition, explained variation in ASD diagnosis. **Conclusions**: These findings suggest that selective and repetitive eating behaviors are characteristic of ASD shape dietary intake, which in turn influences gut microbial diversity. Thus, in humans, the directionality may run primarily from behavior to diet to microbiome, rather than from microbiome to behavior. Our results underscore the importance of incorporating dietary variables into microbiome research and highlight the need for targeted nutritional interventions to improve health outcomes in individuals with ASD.

## 1. Introduction

Autism spectrum disorder (ASD) is a heterogeneous neurodevelopmental condition characterized by deficits in social communication and restricted, repetitive patterns of behavior [1,2]. The prevalence of ASD has risen steadily over the past two decades, with recent estimates indicating that approximately 1 in 31 children in the United States is affected [3], highlighting the critical need for early detection and diagnosis. Beyond its neurobehavioral core symptoms, ASD is frequently accompanied by comorbidities including gastrointestinal (GI) disturbances [4,5], anxiety disorders [6], immune system abnormalities [7] and feeding selectivity [8], all of which contribute substantially to functional impairment and reduced quality of life for affected children and their families. Understanding the biological and environmental factors that shape the heterogeneous presentation of ASD remains a major public health priority [9].

Over the past decade, increasing attention has been directed toward the microbiome–gut–brain axis, a bidirectional communication network linking intestinal microbes with the central nervous system [10,11]. Experimental and translational studies suggest that the gut microbiome may influence neurodevelopment, immune regulation, and behavior through microbial metabolites, immune signaling, and vagal pathways [12,13]. In seminal preclinical work, germ-free mice colonized with microbiome from individuals with ASD exhibited altered behavior and neurochemical profiles, supporting a possible causal link [14]. Subsequent human studies have reported differences in gut microbial composition between individuals with and without ASD, though findings remain inconsistent across cohorts, possibly due to variability in diet, geography, age, and methodological approaches. In a comprehensive review conducted by Qin et al. [15], eight clinical studies were assessed to evaluate the impact of prebiotic and probiotic supplementation in children with ASD. Prebiotics alone led to modest improvements in specific GI complaints, but when paired with an exclusion diet, they were associated with notable reductions in social withdrawal behaviors. Evidence regarding probiotics, however, remains weak, offering little support for their effectiveness in mitigating either GI disturbances or behavioral challenges in this population. Collectively, these findings suggest that, despite encouraging preclinical data, prebiotic and probiotic interventions have yielded only limited clinical benefits for ASD-related GI or behavioral symptoms.

Despite extensive research, key questions remain unresolved. In particular, it is unclear whether observed microbiome alterations in ASD represent causal factors or downstream consequences of dietary preferences, selective eating, or other lifestyle factors commonly associated with the condition. Diet is a major modulator of gut microbial composition and function, yet its contribution to the heterogeneity of microbiome findings in ASD has not been systematically disentangled.

The present study was designed to address this gap. We aimed to examine the relative contribution of dietary patterns and gut microbiome composition to behavioral challenges in children with ASD. By combining detailed dietary assessments with 16S rRNA (16S) and ITS sequencing, we sought to clarify whether behavioral profiles in ASD are more closely associated with dietary factors or with microbiome diversity and composition. We hypothesized that dietary habits, rather than gut microbiome features, would better explain the observed behavioral and functional variability among children with ASD.

## 2. Materials and Methods

### 2.1. Microbiome Study Features

We performed bacteriome and mycobiome analysis on fecal samples collected from a group of 26 children (between 5 and 17 years of age) recruited for this study, including 17 confirmed with an ASD diagnosis and 9 siblings without diagnosis (“SIB”). A total of 27 first-degree family members (parents) without a diagnosis (“PWD”) also participated to this study. Parental samples were included as a within-family reference to contextualize microbial variation, but all primary analyses comparing children with ASD versus siblings were conducted independently of parental data due to differences in age and dietary habits. All subjects participating to this study were exclusively recruited from the Il Cireneo SRL institution (ICS, Vasto, Chieti, Italy) (IRB #: STUDY20230683). The ICS stool samples were collected following a specific protocol and were processed by the same biotechnology company (CosmosID Inc., Germantown, MD, USA) specialized in processing fecal human samples for microbiome analysis. As reported in the literature, studies focused on ASD commonly exhibit sex bias [16,17], so we calculated the percentage of male subjects in each cohort, together with mean age and dietary principal components (PC) obtained from percent energy dietary data subject to centered Log-Ratio transformation (Table 1). Informed consent was obtained from each member participating in the study with an official consent document distributed via hardcopy. Dietary information was also collected using a food diary questionnaire in which participants recorded all foods consumed during seven consecutive days. In addition to dietary intake, the questionnaire obtained data on participants’ sex, age, and reported medication consumption. Identifiable data comprising the participant’s name, email, and telephone number, were collected solely for contact purposes. Completed questionnaires were de-identified upon receipt. The collected data were utilized to characterize the gut microbiome composition, comprising bacteriome and mycobiome, of the participants. The overall study design is illustrated in Figure 1.

### 2.2. Sample Size and Study Design

This study was conducted as an exploratory pilot investigation to assess the relative contributions of dietary patterns and gut microbiome composition in children with ASD. Participant recruitment was limited to a single institution (ICS), resulting in a total of 26 children (17 with ASD, 9 non-ASD siblings) and 27 parents without ASD. Given the high interindividual variability commonly observed in microbiome analyses, this sample size was not determined by formal power calculations but was chosen to provide preliminary data for hypothesis generation, methodological feasibility, and effect size estimation. The within-family design (ASD children, siblings, and parents) helped reduce variability and strengthen comparative analyses. Findings from this pilot cohort will inform sample size calculations and study design for future adequately powered investigations. Rather than recruiting an independent neurotypical cohort, we employed a within-family control design, which helped control for genetic background and shared environmental factors, including household dietary habits, thereby reducing interindividual variability in microbiome composition. While this approach strengthens internal validity, we acknowledge that the absence of an independent neurotypical control group may limit generalizability. Future studies with larger, more diverse cohorts should include age- and sex-matched neurotypical participants to validate and extend these findings.

### 2.3. Fecal Samples for Microbiome Analysis

Stool samples from children with autism, their non-autistic siblings, and parents (*n* = 53) were collected between 22 January 2024, and 20 September 2024, and subjected to multi-kingdom microbiome sequencing, encompassing both bacteriome and mycobiome. Upon arrival at the CosmosID laboratory, samples were stored at −80 °C. After completion of sample collection, all specimens were processed and sequenced in a single batch approximately one month later, ensuring consistency across the dataset.

### 2.4. DNA Extraction

Metagenomic DNA was extracted from fecal aliquots thawed on ice and resuspended in 600 mL DNA stabilization buffer (Stratec Biomedical, Birkenfeld, Germany) and 400 mL phenol/chloroform/isoamyl alcohol (25:24:1, by vol; Sigma-Aldrich, St. Louis, MO, USA). Cells were mechanically lysed (3 × 6.5 m/s for 40 s) with 500 mg 0.1 mm glass beads (Roth) using a bead-beater (MP Biomedicals) fitted with a cooling adapter. After heat treatment (95 °C, 8 min) and centrifugation (16,000 3× *g*; 5 min; 4 °C), 150 mL supernatant was incubated with 15 mL ribonuclease (0.1 mg/mL; Amresco, Solon, OH, USA) at 37 °C and centrifuged (5503× *g*; 30 min). DNA was purified with the NucleoSpin gDNA Clean-up Kit (Macherey-Nagel, Düren, Germany), following the manufacturer’s instructions. Concentrations and purity were determined with the (Wilmington, DE, USA) and Qubit system (Thermo Fisher Scientific, Oakwood, OH, USA) and stored at −20 °C.

### 2.5. Targeted Amplification

Amplifications of the V3–V4 Region (for 16S) and the 5.8S rRNA genes (for ITS) were obtained utilizing 16S-515 (5′-GGA CTA CCA GGG TAT CTA ATC CTG-3′) and 16S-804 (5′-TCC TAC GGG AGG CAG CAG T-3′) and ITS1 (5′-(TCC GTA GGT GAA CCT GCG G-3′) and ITS4 (5′-TCC TCC GCT TAT TGA TAT GC-3′) primers. PCR was performed using Q5 High-Fidelity Master Mix (New England Biolabs) with primers at 0.05 μL/mM and 100 ng of template DNA per 50 μL reaction. Amplification on a Biometra Tadvanced thermocycler (Analytik Jena, Jena, Germany) included an initial denaturation at 98 °C for 3 min, 30 cycles of denaturation (98 °C, 10 s), annealing (16S: 55 °C, 10 s; ITS: 58 °C, 20 s), extension (72 °C, 10 s), and a final extension at 72 °C for 3 min.

### 2.6. Library Generation and Sequencing Workflow

Amplicon libraries were purified and uniquely barcoded before performing emulsion PCR according to the Ion Torrent S5 Prime workflow (Thermo Fisher Scientific). Equal volumes of bacterial 16S rRNA and fungal ITS amplicons were combined and cleaned using AMPure XP beads (Beckman Coulter, Hebron, KY, USA) to remove residual primers. The pooled amplicons were then treated with an end repair enzyme for 20 min at room temperature, followed by a second AMPure cleanup. Ligation of the Ion Torrent P1 adapter and a unique barcoded ‘A’ adaptor was performed at 25 °C for 30 min. All barcoded samples were subsequently pooled in equal amounts (10 μL each) and size-selected for the expected 16S and ITS fragment range (200–800 bp) using Pippin Prep (Sage Biosciences, Beverly, MA, USA). The library underwent seven cycles of amplification and was quantified on a StepOne qPCR instrument (Thermo Fisher Scientific) prior to dilution to 100 pM for IonSphere templating on the Ion Chef. Sequencing was carried out on an Ion Torrent S5 platform, and barcode-separated reads were processed through a custom pipeline utilizing the Greengenes V13_8 and UNITE V7.2 databases for taxonomic classification of 16S rRNA and ITS sequences, respectively.

### 2.7. Microbiome Data Processing

Microbiome data were analyzed using the CosmosID-QIIME2 integrated pipeline through the Cosmos-Hub cloud platform, which incorporates the QIIME2 framework for operational taxonomic unit (OTU) clustering and taxonomic assignment. Sequences were clustered into OTUs at 97% similarity, and taxonomic classification was performed using the SILVA v138 reference database. All samples underwent rigorous quality control: low-quality and chimeric reads were removed, and OTUs/ASVs representing <0.01% of total reads or detected in <10% of samples were excluded to minimize noise. Samples with fewer than 10,000 reads were removed prior to downstream analyses. The resulting abundance table was rarefied to 10,000 reads per sample and normalized by relative abundance. Negative controls (blank DNA extractions) were included in each sequencing batch to monitor potential contamination; no significant microbial signal was detected. α-diversity (Shannon and Simpson) and β-diversity (Bray–Curtis and weighted UniFrac) metrics were calculated on the filtered dataset. These procedures ensured high-quality, reliable data suitable for downstream diversity and compositional analyses.

### 2.8. Dietary Data

Dietary information for ASD, SIB and PWD participants was obtained through an Italian dietary survey completed by parents, who recorded a 7-day food diary documenting all foods and beverages consumed at breakfast, lunch, dinner, and snack occasions, allowing for detailed assessment of dietary patterns and comprehensive reporting of daily intake. Food-level intake data were available for 79 individuals: ASD (*n* = 26), SIB (*n* = 12), PWD (*n* = 41). The dietary survey captured frequencies of consumption for 112 distinct food items, which were subsequently aggregated into eight predefined food groups: dairy, fast food/ultra-processed, fruits, grains, proteins, sugar-sweetened beverages, sweets and sugary foods, and vegetables. Food-level intake data were used to examine dietary diversity among the three study cohorts, with the aim of identifying groups that exhibited significantly different intake patterns. The 7-day food diary was used to record the frequency of consumption for each specific food item across all eating occasions (breakfast, lunch, dinner, and snacks). To standardize the data, raw frequencies were converted into percentages of each participant’s total weekly food consumption, generating relative abundance values for each food item. These values were then used to calculate dietary diversity indices. Statistical analyses were conducted using the Kruskal–Wallis test to assess overall group differences. When significant effects were detected, post hoc pairwise comparisons were performed to identify specific cohort differences.

### 2.9. Statistical Analysis

All statistical analyses were conducted in RStudio (version 2024.09.1+394). Changes in genus- and species-level abundance were evaluated using non-parametric multivariate methods, specifically the Kruskal–Wallis test. Microbial diversity was assessed with the Shannon index, which accounts for both abundance and distribution, while richness was calculated to reflect the total counts of bacterial and fungal taxa within each sample. Associations between microbial communities and outcomes were analyzed using the Adonis function from the R package vegan (v2.7-1) for non-parametric multivariate distance-based testing. For continuous and binary outcomes, the Wilcoxon rank-sum test was applied. Longitudinal comparisons were performed with pairwise Multiple Comparison of Mean Ranks using the Kruskal–Wallis test followed by Bonferroni–Dunn post hoc adjustment (stats package, v4.5.1). Effect sizes (η^2^) were calculated to provide a measure of the magnitude of group differences, and *p* values were adjusted for multiple testing using the Benjamini–Hochberg false discovery rate (FDR) procedure. In addition to the non-parametric tests (Kruskal–Wallis and Wilcoxon), we conducted supplementary analyses using LMMs to account for the hierarchical structure of the data, treating “Family” as a random effect. These models were applied separately to bacterial (16S), fungal (ITS) and food datasets. However, given that only 14 families were available for 16S and ITS and 21 available for food analysis, the number of independent clusters was insufficient to detect significant effects with adequate statistical power. Consequently, the LMM analyses did not yield significant differences between groups, likely reflecting limited sample size rather than the absence of group-level effects (Appendix A).

### 2.10. Reporting Guidelines

This study adhered to the Strengthening the Organization and Reporting of Microbiome Studies (STORMS) guidelines to ensure methodological rigor, transparency, and reproducibility in microbiome research. All key elements related to study design, sequencing, data processing, and analysis were reported in accordance with the checklist proposed by Mirzayi et al. [18]. A completed STORMS checklist is provided as Appendix A.

### 2.11. Ethics Approval and Consent to Participate

All participants provided informed consent to be included within this study (Institutional Review Board #: STUDY20230683).

## 3. Results

### 3.1. Bacteriome Analysis Revealed No Meaningful Correlation with ASD Diagnosis

First, we evaluated how much of the phenotypic variation in ASD diagnosis was attributable to the microbiome. The characterization of the microbiome through 16S analysis indicated non-significant correlation between β-diversity and ASD diagnosis (Figure 2A) (PC1: 45.2%; PC2: 4.4%; PC3:3.8%). Moreover, α-diversity, measured by the Shannon index, did not differ significantly among ASD, SIB, and PWD groups (pairwise Wilcoxon tests, all *p* > 0.05) (Figure 2B). Finally, analysis of the heatmap representing the relative abundance of bacterial genera/species across different cohorts revealed negligible variability in taxa composition among the groups (z-scores represented in Figure 2C). The permutational multivariate analysis of variance (PERMANOVA) revealed no significant differences in β-diversity among ASD, SIB, and PWD cohorts (adonis2, R^2^ = 0.045, FDR > 0.05). Pairwise comparisons (ASD vs. SIB, ASD vs. PWD, SIB vs. PWD) also showed no significant group-level separation. These results indicate that gut microbial community structures were largely comparable across cohorts (Table 2).

### 3.2. Mycobiome Profile Showed No Detectable Differences Between ASD and Non-ASD Individuals

Next, since we did not find any significant differences in relation to bacterial populations, we analyzed the fecal samples focusing on the fungal community. The mycobiome characterization through ITS analysis indicated also non-significant correlation between β-diversity and ASD diagnosis (Figure 3A) (PC1: 40.5%; PC2: 4.6%: PC3:3.4%). Similarly, α-diversity did not differ significantly among ASD, SIB, and PWD groups (pairwise Wilcoxon tests, all *p* > 0.05) (Figure 3B). Lastly, heatmap analysis representing the relative abundance of fungal species across different cohorts revealed negligible differences in taxa composition among the groups, with the Kruskal–Wallis test per taxon across the three groups indicating statistically significant differences only for *Aspergillus candidus* and *Aspergillus tritici* (*p* < 0.05), although pairwise comparisons did not highlight any difference between ASD and SIB groups, but only difference in comparison with the PWD cohort (z-scores represented in Figure 3C). The PERMANOVA analysis of ITS profiles revealed no significant β-diversity differences among ASD, SIB, and PWD (adonis2, R^2^ = 0.043, FDR > 0.05). Pairwise comparisons (ASD vs. SIB, ASD vs. PWD and SIB vs. PWD) similarly showed no significant differences among the three cohorts. These results indicate that, consistent with the previously mentioned bacterial findings, fungal community structures were broadly similar across all participants, suggesting negligible taxonomic diversities within family clusters (Table 3).

### 3.3. Patterns of Dietary Intake Were Consistently Related to ASD

As we found negligible microbiome differences in relation to ASD diagnosis, we next assessed potential associations between dietary patterns and the autistic phenotype. We examined the food diaries of the three study groups and identified statistically significant differences in dietary intake patterns among individuals with ASD, their siblings without ASD, and their parents without ASD. The observed differences were related to: macronutrient distribution (e.g., relative proportions of carbohydrates, proteins, and fats); consumption frequency of specific food groups (such as dairy, grains, fruits, and vegetables); intake of processed and high-sugar foods; dietary diversity and variety across meals. Analysis of dietary intake revealed that the most pronounced differences among the groups were observed in two food categories. First, the ASD group consumed significantly higher amounts of sweet and sugary foods compared to both the SIB and PWD cohorts (Kruskal–Wallis test, *p* < 0.05). Second, vegetable intake was significantly smaller in the ASD group relative to SIB and PWD (Kruskal–Wallis test, *p* < 0.05). In contrast, no statistically significant differences were detected between the SIB and PWD groups for either of these categories, suggesting that the observed patterns were specific to individuals with ASD (Kruskal–Wallis test, *p* = ns). These findings indicate that particular dietary habits may be characteristic of the autistic phenotype and could contribute to nutritional or metabolic profiles associated with ASD.

The aforementioned differences are reported in the heatmap (Figure 4A), which illustrates the distribution of dietary intake, indicating the percentage of each food category consumed by the three groups during a seven-day period. Pairwise comparisons are reported in Figure 4B, showing the Mean Relative Abundance (%) of each food category consumed by each group. Additional differences were observed for fast food/ultra-processed items and fruits (Kruskal–Wallis test, *p* < 0.05); these differences were present when comparing ASD and SIB groups to the PWD cohort, but no significant differences were detected between ASD and SIB (Benjamini–Hochberg–adjusted pairwise comparisons = ns). This suggests that while these patterns distinguish parents from the younger generations, they are unlikely to be specifically associated with ASD. Collectively, these findings highlight that particular dietary habits may be characteristic of the autistic phenotype, reflecting preferences or behaviors that could influence overall nutritional intake and metabolic profiles in individuals with ASD.

## 4. Discussion

A growing and increasingly nuanced body of research has examined the association between the gut microbiome and ASD, exploring whether variations in microbial community composition are linked to neurodevelopmental outcomes [19,20,21,22]. While several bacterial taxa have been repeatedly implicated across studies, findings remain inconsistent and often contradictory. For example, a meta-analysis of nine independent studies reported that children with ASD exhibited reduced relative abundances of *Bacteroides* and *Bifidobacterium*, alongside higher levels of *Faecalibacterium* and *Lactobacillus*, compared with neurotypical controls [23]. Conversely, Ahmed et al. [24] observed increased *Bacteroides* levels in ASD, underscoring the lack of consensus regarding specific microbial signatures. These discrepancies likely reflect heterogeneity within the ASD population and the influence of confounding factors such as diet, age, geography, and methodological variability. Despite these inconsistencies, the broader literature supports a complex, context-dependent relationship between gut microbial diversity and ASD-related characteristics [25].

### 4.1. Microbiome Findings and Fungal Signatures

In our multi-kingdom metagenomic analysis, we found minimal evidence for a direct association between either bacterial or fungal composition and ASD diagnosis. Among the few differences observed, *Aspergillus* species—particularly *A. candidus* and *A. tritici*—were detected at different relative abundances among children with ASD and their siblings compared with parents. Although these fungi are not typically considered major components of the human gut, their detection suggests potential environmental or dietary origins. *Aspergillus* species are known to produce metabolites such as gliotoxins, which have been associated with inflammatory and immunomodulatory processes [26]. A recent work by Baker et al. [27] reported that metabolites specifically produced by *Aspergillus* species, such as furan-2,5-dicarboxylic acid and 5-hydroxymethyl-2-furoic acid, were elevated in ASD children compared to neurotypical controls. Notably, antifungal treatment with itraconazole led to a reduction in these metabolites and a concurrent recovery from ASD symptoms. Given that fungal metabolites, including mycotoxins, possess neurotoxic potential, altered gut mycobiome composition may indirectly affect brain function via immune or metabolic pathways [28]. Supporting this hypothesis, Markova [29] found that nearly all the autistic children examined, along with their mothers, exhibited filamentous *Aspergillus* fungi in their blood. This suggests possible vertical transmission from mother to child before birth, implying that colonization may occur early in development and potentially contribute to neurodevelopmental alterations.

### 4.2. Microbiome Variation and Dietary Influence

Across our analyses, α-diversity and microbiome–diet association indices showed only weak relationships with clinical status, consistent with prior findings [15,30]. This pattern reinforces that microbial compositional variation among ASD, sibling (SIB), and parental (PWD) groups was limited. The small microbial differences, combined with stronger dietary associations, suggest that dietary intake patterns represented the most prominent group-level distinction in this cohort. This aligns with methodological evidence indicating that small-sample microbiome studies are prone to inflated false-positive rates, and that multiple-comparison corrections frequently remove marginal or non-reproducible signals [31,32].

### 4.3. Behavioral–Dietary Pathways and Clinical Implications

Children with ASD in our cohort consumed fewer vegetables, greater amounts of sweets and sugary foods, and exhibited reduced dietary diversity compared with their non-ASD family members. These observations are consistent with previous reports describing selective food preferences and restricted eating behaviors among autistic individuals [33,34,35]. Such food selectivity is clinically relevant, as it is associated with avoidant/restrictive food intake disorder (ARFID) and potential nutrient insufficiencies requiring medical attention [36].

The lower dietary quality observed among ASD participants emphasizes the importance of considering targeted nutritional strategies, as greater dietary diversity has been linked to improved overall health and cognitive outcomes [37]. Although prior studies have proposed that dietary composition may influence microbial metabolism and host physiology via neuroactive metabolites such as short-chain fatty acids (SCFA)s and tryptophan derivatives [38,39,40,41], the present study did not measure these metabolites; thus, these relationships remain hypothetical.

### 4.4. Methodological Considerations and Study Limitations

Several limitations should be acknowledged.

First, the sample size—particularly for ASD (*n* = 17) and SIB (*n* = 9)—was modest, limiting power to detect subtle microbial differences. No a priori power analysis was conducted, as the study used a convenience sample recruited from a single institution. While this study identified group-level differences in dietary patterns, no corresponding microbiome alterations were observed. However, given the modest sample size and cross-sectional design, these results should be interpreted as exploratory and associative. The absence of microbiome differences may reflect limited statistical power rather than a true lack of effect. Future studies employing larger, longitudinal designs are required to disentangle the directionality of the relationship between behavior, diet, and microbiome composition.

Second, inclusion of parental controls (PWD) introduces potential confounding effects due to age and diet differences. The inclusion of PWD as a comparator group provided a broader familial and environmental context for interpreting microbiome and dietary variation within households and allowed us to account for shared genetic background, living environment, and household dietary practices, offering valuable insight into intergenerational differences. Moreover, the use of parental controls is supported by family-based microbiome studies demonstrating that shared genetics, diet and household environment contribute significantly to microbiome composition [42,43]. However, given the natural differences between adults and children in both gut microbiome composition and dietary patterns, comparisons involving PWD were interpreted with caution. For this reason, our primary analyses focused on comparisons between ASD children and their non-ASD siblings, thereby minimizing these confounding effects while maintaining a controlled familial framework.

Third, dietary data were analyzed at the food-group level, without nutrient-level granularity (e.g., fiber, macronutrient, or micronutrient content). Although 7-day food diaries provide robust coverage of habitual intake, they remain subject to recall and reporting errors.

Fourth, the absence of metabolomic data (e.g., SCFAs, bile acids) prevents functional interpretation of microbiome–diet relationships. This limitation, driven by cost constraints, should be addressed in future integrated multi-omics studies.

Finally, although some food-group differences were significant in non-parametric tests, they were not retained in the LMMs—likely reflecting the limited number of family clusters (*n* = 21), which constrained model power and precluded adjustment for all confounders (e.g., BMI, medication use, socioeconomic status). Small hierarchical samples are known to reduce the stability of LMM estimates [44].

Despite these limitations, this pilot study provides valuable preliminary insight by integrating multi-kingdom microbiome profiling with detailed dietary data in a within-family design. The results highlight strong behavioral and dietary associations with ASD, while showing negligible direct microbiome effects. These findings emphasize the need for larger, adequately powered studies including independent neurotypical controls and nutrient-level dietary data.

## 5. Conclusions

In the present study, dietary intake patterns showed stronger group-level associations with ASD status than microbiome composition. Specifically, children with ASD demonstrated lower consumption of vegetables and higher intake of sweets and sugary foods compared with non-ASD family members, while bacterial and fungal community profiles exhibited minimal and statistically non-significant differences after correction for multiple testing. These findings suggest that, within this cohort, dietary variation contributed more substantially to group differentiation than microbial composition. However, given the modest sample size and cross-sectional design, these associations should be interpreted as descriptive rather than causal. Future research integrating microbiome, metabolomic, and inflammatory analyses will be essential to clarify the complex interrelationships between behavior, diet, and gut ecology in ASD and to evaluate the potential for dietary interventions to support overall health in this population.

## Figures and Tables

**Figure 1 nutrients-17-03476-f001:**
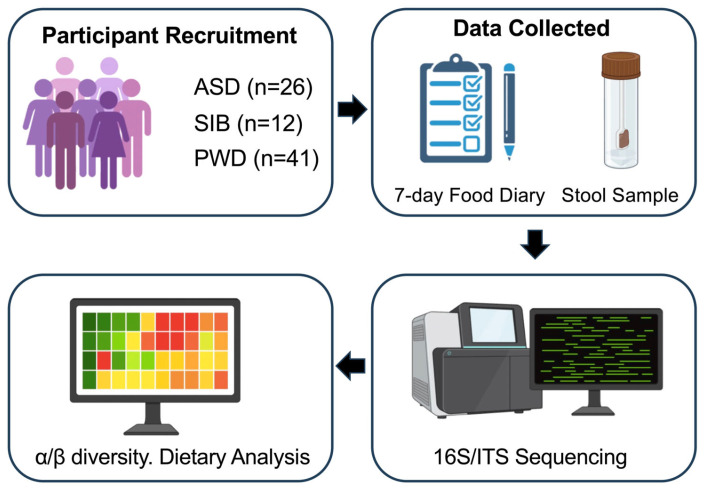
Overview of the study workflow. The figure summarizes participant recruitment from the ICS institution (ASD, SIB, and PWD), collection of 7-day food diaries and stool samples, 16S rRNA and ITS sequencing, bioinformatics processing, and downstream statistical analyses including α- and β-diversity, PERMANOVA, and Linear Mixed-Effects Models (LMMs).

**Figure 2 nutrients-17-03476-f002:**
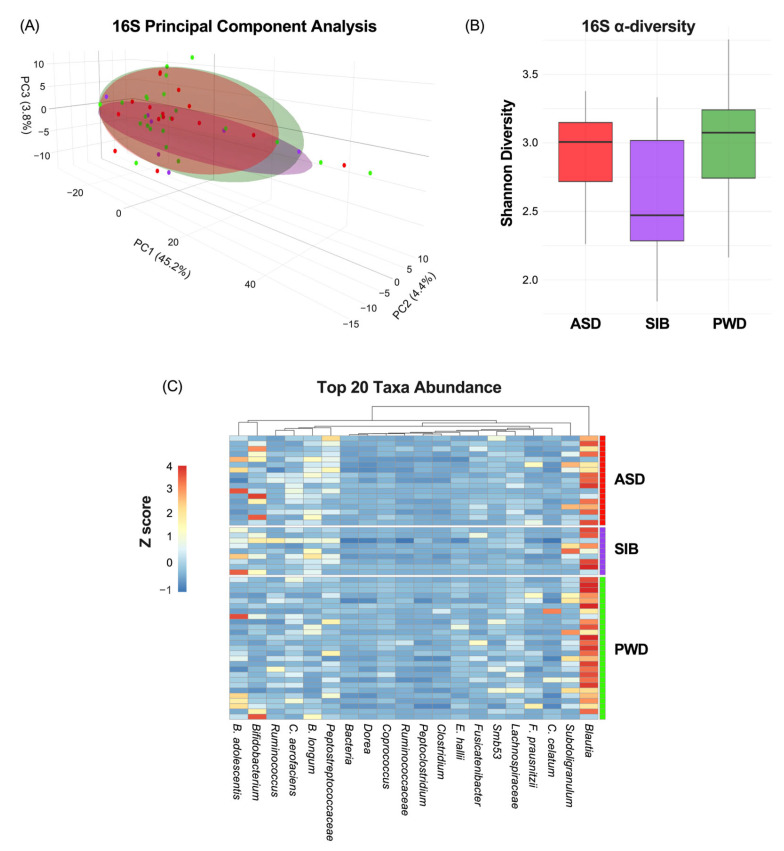
Microbiome variation across ASD, SIB, and PWD groups. (**A**) PCA of 16S profiles showed no significant association between β-diversity and ASD diagnosis (PC1: 45.2%; PC2: 4.4%; PC3: 3.8%). (**B**) Shannon index α-diversity did not differ among groups (pairwise Wilcoxon tests, all *p* > 0.05). (**C**) Heatmap analysis revealed negligible variability in taxa composition among the groups.

**Figure 3 nutrients-17-03476-f003:**
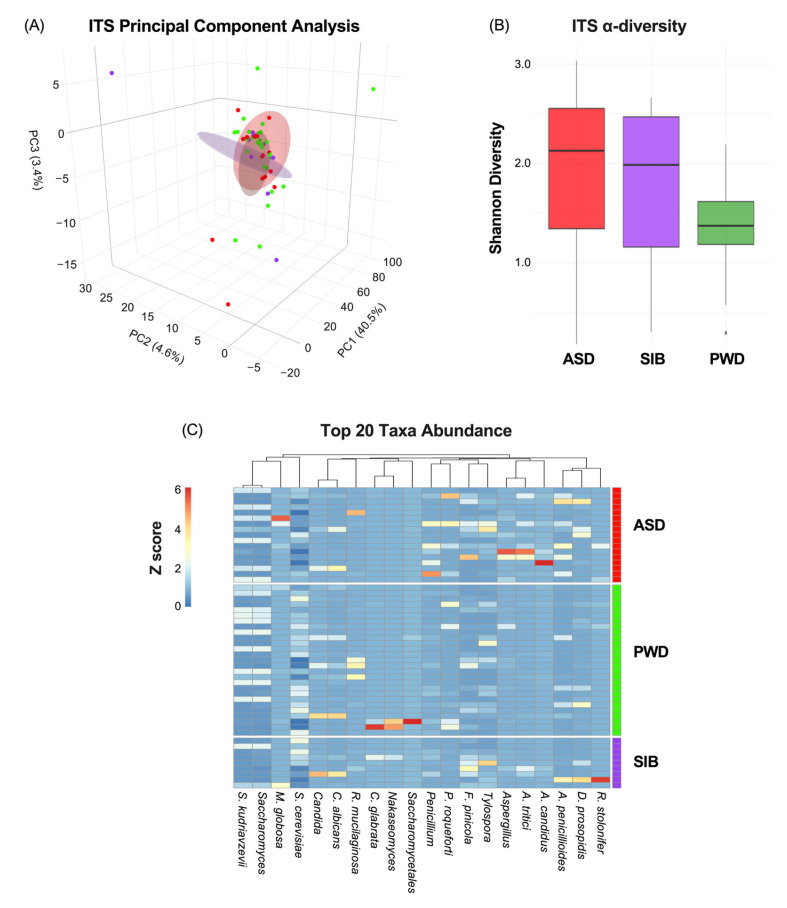
Mycobiome diversity and taxonomic composition across ASD, SIB, and PWD groups. (**A**) PCA of ITS profiles showed no significant association between β-diversity and ASD diagnosis (PC1: 40.5%; PC2: 4.6%; PC3: 3.4%). (**B**) α-diversity, assessed by the Shannon index, did not differ significantly among groups (pairwise Wilcoxon tests, all *p* > 0.05). (**C**) Heatmap of relative fungal abundances revealed minimal compositional differences, with Kruskal–Wallis tests identifying significance only for *Aspergillus candidus* and *Aspergillus tritici* (*p* < 0.05). Pairwise Dunn’s tests showed that these differences were driven by comparisons with the PWD cohort, but not between ASD and SIB groups.

**Figure 4 nutrients-17-03476-f004:**
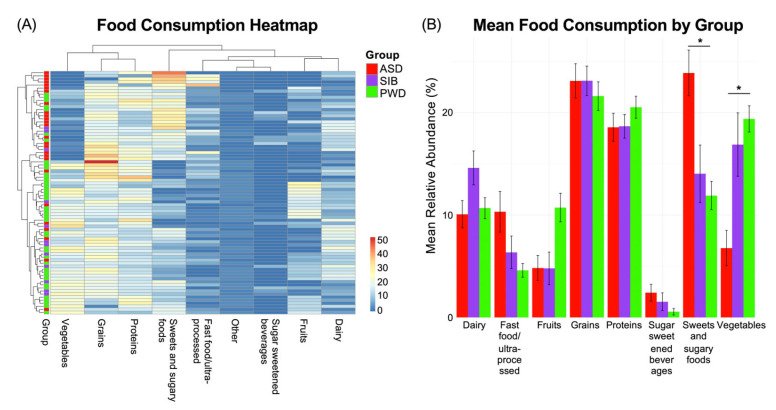
Dietary intake patterns across ASD, SIB, and PWD groups. (**A**) Heatmap of food diary data (% consumption over seven days) showed significant differences in dietary intake among groups. Individuals with ASD consumed more sweet/sugary foods and vegetables than both SIB and PWD (Kruskal–Wallis, *p* < 0.05), while no differences were detected between SIB and PWD. (**B**) Pairwise comparisons also revealed differences in fast food/ultra-processed items and fruits, distinguishing ASD and SIB from PWD but not from each other; * = *p* < 0.05.

**Table 1 nutrients-17-03476-t001:** The ‘Phenotype’ row summarizes the comparison among ASD, SIB, and PWD groups used in the microbiome and dietary analyses. Subsequent rows provide subject characteristics, including mean age, percentage of males, dietary principal components (calculated from Centered Log-Ratio transformed percent energy intake), and the dietary Shannon diversity index.

	Microbiome Analysis			Dietary Analysis		
Phenotype	ASD	SIB	PWD	ASD	SIB	PWD
*n*	17	9	27	26	12	41
Age	9.1	10.9	45.9	9	11.7	45.4
Male%	82.4	66.7	44.4	84.6	66.7	48.8
Dietary PC1				−1.02	0	1.08
Dietary PC2				−2.72	9.37	−1.19
Dietary PC3				−3.42	−6.01	−1.57
Dietary Shannon Index				1.62 ± 0.2	1.73 ± 0.1	1.68 ± 0.2

**Table 2 nutrients-17-03476-t002:** PERMANOVA results on bacterial β-diversity, based on Bray–Curtis dissimilarities, 999 permutations (adonis2). FDR correction by the Benjamini–Hochberg procedure.

Comparison/Variable	F-Model	R^2^	*p* Value	FDR	Interpretation
**Overall PERMANOVA**					
Group (ASD, SIB, PWD)	1.33	0.045	0.19	0.27	No Significant
Age	0.84	0.007	0.47	-	No Significant
Sex	0.62	0.002	0.63	-	No Significant
**Pairwise comparisons**					
ASD vs. SIB	1.12	0.041	0.31	0.47	No Significant
ASD vs. PWD	1.28	0.052	0.17	0.26	No Significant
SIB vs. PWD	0.98	0.036	0.45	0.61	No Significant

**Table 3 nutrients-17-03476-t003:** PERMANOVA results on fungal β-diversity based on Bray–Curtis dissimilarities (adonis2), 999 permutations. FDR correction by the Benjamini–Hochberg procedure.

Comparison/Variable	F-Model		R^2^	*p* Value	FDR	Interpretation
**Overall PERMANOVA**						
Group (ASD, SIB, PWD)	1.21		0.043	0.23	0.35	No Significant
Age	0.79		0.006	0.51	-	No Significant
Sex	0.67		0.003	0.58	-	No Significant
**Pairwise comparisons**						
ASD vs. SIB	1.05		0.038	0.37	0.51	No Significant
ASD vs. PWD	1.18		0.046	0.25	0.38	No Significant
SIB vs. PWD	0.93		0.032	0.43	0.59	No Significant

## Data Availability

The datasets supporting the conclusions of this study are available from the corresponding author, Luca Di Martino, upon reasonable request. Data will be securely retained for a minimum of five years within Box, the cloud-based storage platform authorized by Case Western Reserve University for the management of restricted data. This service ensures secure, long-term storage, controlled access, and reliable file sharing in compliance with institutional data governance policies.

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
