# Peer review of "Dietary Patterns, Not Gut Microbiome Composition, Are Associated with Behavioral Challenges in Children with Autism: An Observational Study"

_nutrients, 2025, doi:10.3390/nu17213476_

Round 1

Reviewer 1 Report

Comments and Suggestions for Authors
  1. The current title overstates causal inference (please avoid causal phrasing like “underlie challenges”) and omits the study type (pilot, observational study).
  2. The introduction requires better paragraphing and a more coherent structure. Suggest to first cover the epidemiology and clinical relevance of ASD, then the hypothesized role of the gut-brain axis and recent evidence (citation: pubmed.ncbi.nlm.nih.gov/31083360), key unresolved issues, study rationale/hypotheses.
  3. The study objectives are implied but never explicitly stated anywhere.
  4. How was the sample size determined? The sample is considerably small, particularly for microbiome analyses where interindividual variability is large.
  5. There is no independent neurotypical control group. Why?
  6. In this study, parents are included in microbiome comparisons despite age and diet (known confounding factors) being drastically different. Why?
  7. The Methods section should clearly specify data preprocessing steps (e.g., quality filtering thresholds, read counts, normalization etc).
  8. The statistical analysis plan is problematic as the use of only the Kruskal–Wallis and Wilcoxon tests ignores the hierarchical data structure (families as clusters). This leads to pseudoreplication and potentially inflated significance.
  9. The small number of significant taxa (e.g., Bifidobacterium longum, Eubacterium hallii) without correction for multiple testing makes the results prone to false positives. 
  10. The study claims that “dietary patterns explained variation in ASD diagnosis,” yet no multivariate model adjusting for potential confounders (age, sex, medication use, BMI, antibiotic exposure, stool consistency, socioeconomic status, or ASD severity) is presented. Why?
  11. The study uses Ion Torrent S5 with 16S V3–V4 and ITS1–ITS4 sequencing, analyzed via Qiime 1.8, which is outdated and unsupported for a while now. Modern microbiome studies use Qiime 2, DADA2, or Deblur with amplicon sequence variants (ASVs) for greater resolution.
  12. No visualization (PERMANOVA, NMDS, or compositional analyses like ANCOM, ALDEx2) is reported.
  13. Quality-control metrics (read depth, rarefaction, contamination checks, negative controls) are not reported.
  14. Discussion of study limitations inadequate. No nutrient-level analysis (macronutrient distribution, fiber intake, micronutrient deficiencies) is provided. Thus, mechanistic links to microbiome composition or health cannot be inferred. Moreover, the 7-day food diary method is prone to recall and reporting errors.
  15. The conclusion is speculative and unsupported by cross-sectional data. No temporal or causal inference can be drawn. Please temper the conclusions accordingly.
  16. The current PCA and heatmap figures are visually cluttered, low in resolution, and can be improved in terms of effectiveness.
  17. Consider adding a schematic of the study flow (e.g., participant recruitment, data collection, analysis pipeline).
  18. To ensure rigor and transparency, this study should be reported according to microbiome-specific extensions such as STORMS (STrengthening the Organization and Reporting of Microbiome Studies). A copy of the completed checklist should be appended for review. 
Comments on the Quality of English Language

Moderate edits required.

Reviewer 2 Report

Comments and Suggestions for Authors

This is a well conducted study reported in a well written manuscript. The principle finding is that children with autism spectrum disorder tend to follow unhealthy diets, with a paucity of vegetables and excessive sweets. The authors reasonably conclude that the gut microbiome is affected in autism, but that there is no causal pathway for dysbiosis to autism. Instead, the affected children may disrupt their own gut microbiomes by electing substandard diets. One possibility that might bear inclusion in the discussion: is it possible that the ill-advised choice of food causes changes in the microbiome that, in turn, lower the threshold for secondary psychiatric conditions such as depression?

Reviewer 3 Report

Comments and Suggestions for Authors

The current study entitled “Dietary Patterns, Not Gut Microbiome, Underlie Challenges in Children with Autism” clearly reflects the study design and findings and naturally created interest from the reviewer, given the large amount of interest from the general public in this topic. The reviewer commends the research team on a relatively well-designed and timely study on this particular topic. The team have presented a study that evaluates the effect of the gut–brain axis in ASD and whether the microbiota (determined by measuring the microbiome) is a primary driver in the development of ASD but assessing not only the microbiome of the ASD children, but their siblings and parents. The authors present a compelling case that dietary behaviours, rather than microbiota composition are more strongly associated with ASD. Further, the inclusion of both bacteriome and mycobiome profiling is appreciated and is a strength of the study. However, there are some comments that this reviewer wishes to raise:

  • One limitation to this study is that it focused on microbiome data only and did not evaluates metabolic changes associated with an altered microbiota (i.e. SCFAs). Could the research team outline why this was not assessed in this study? If it were due to cost, then this is understandable, but this should be listed as a major limitation of the study as it partly shows a functional aspect of the microbiota compositions that was not assessed but has impact on the gut-brain axis as reported by numerous studies in this field.
  • The introduction and discussion are present both as lone long block of text, which negatively impacts the flow of these sections. Please break up these sections into logical segments that guide the reader through the background, rationale, and aims of the study, as well as the interpretation of the data in the discussion. On this note, there are also some sections in the methods that contains inconsistent verb tense, with several sentences written in present or future tense (e.g., “is” instead of “was”). This needs to be corrected throughout, as scientific writing should describe completed procedures in the past tense (good example of this is the Data Extraction sub section). The results and figures also have some inconsistencies in terminology that needs to be tightened to ensure that it is then consistent throughout the manuscript (e.g., “PWD” vs. “parental controls”),
  • The reviewer understands that assessing the microbiota composition via microbiome is expensive and is often the limiting factor to recruiting and analysing a good number of samples. However, the sample size is relatively small, which may limit the power to detect subtle microbiome differences. It appears the manuscript did not indicate whether a power analysis was conducted to determine the sample size. Given the relatively small cohort, especially in the ASD and sibling groups, the authors should clarify whether the study was adequately powered to detect meaningful differences in microbiome or dietary variables. If not, this limitation should be acknowledged in the discussion. If this was purely a convenience sample size then this should be noted as well. This is of importance, because the lack of significant findings may be due to the study being underpowered rather than a true absence of effect. Given the findings this is probably unlikely, because the statistical approach is appropriate, but it is something that the authors need to address in the manuscript to demonstrate caution and not oversell the findings.
  • Some of the microbiome analysis relies on OTU clustering and something that many studies have now advanced further with using other means instead of Qiime 1.8. It would be beneficial to add 1 – 2 lines in the methods about why this method was selected for given the field’s advancements (preferred). Alternatively, 1-2 lines could be added in the discussion outlining this method selection as a limitation as this may have meant that finer taxonomic or functional differences between the groups may have been overlooked. This would appease those who utilise more advanced metagenomic methods in the field to acknowledge this potential limitation. Given the data presented the reviewer has suggested that 1-2 lines could be added in the methods would be the preferred option as noting this as a limitation may detract from the data presented which is quite robust.
  • In the discussion, there is also limited mention of the fungal data, particularly the Aspergillus candidus and tritici differences. The manuscript does not really highlight or indicate the relevance of these differences in ASD despite finding and reporting these results (i.e. please interpret the results for completion). A short discussion of potential immunological or neurotoxic implications (if any) would strengthen this section. This study: https://pmc.ncbi.nlm.nih.gov/articles/PMC7572136/#:~:text=In%20this%20study%20organic%20acid,not%20attempted%20in%20this%20study. May be of use given its findings in ASD Px and this Mothers and the impact treatment had on these microbes.
  • The discussion could also benefit from discussing how do sensory sensitivities and restricted interests in ASD influence food selectivity? Could altered dietary intake affect neuroinflammation or neurotransmitter synthesis via microbial metabolites? These links are important and would help contextualise the findings (i.e. the team have are aiming to report ASD → behaviour → diet → microbiome instead of the other way around as some studies have suggested. Hence it is important to address this concept clearly and also ensure that the findings of the diet are highlighted and contextualised to ensure that the mechanism proposed by the research team is clear. This also goes back to the reviewer’s first comment around the lack of SCFA analysis. With regard to inflammation, it would have been at least beneficial to perform some biomarker analysis related to inflammation even if only a plasma LPS concentration given the negative this has on the BBB and both systemic and neuroinflammation as previously described in numerous studies via the innate immune system being stimulated over a prolonged period. If LPS did not change then this would further support the proposed mechanism, but alas, this was not measured. The authors would need to note this as a limitation (i.e. no inflammatory markers measured, etc) and that this should be a future direction to consolidate these findings and support the proposed mechanism presented in this manuscript.

Round 2

Reviewer 1 Report

Comments and Suggestions for Authors
  1. I would appreciate greater justification for the use of PWD as a comparator in this study especially given differences between adults and children in terms of natural gut microbiome and diet.
  2. Reported “significant” taxa (e.g., Bifidobacterium longum) did not survive FDR correction, yet authors write "significant differences”. Please remove any mention of nominal significance unless adjusted p < 0.05.
  3. It is highly premature to say that directionality may run primarily from behavior --> diet --> microbiome. Fundamentally, the study cannot support its headline conclusion that diet, not microbiome, is associated with ASD. The correct interpretation is insufficient statistical power to detect realistic effect sizes.
Comments on the Quality of English Language

Moderate edits required.
